

# Subset-based stereo calibration method optimizing triangulation accuracy

Oleksandr Semeniuta

Department of Manufacturing and Civil Engineering, Norwegian University of Science and Technology, Gjøvik, Norway

## ABSTRACT

Calibration of vision systems is essential for performing measurement in real world coordinates. For stereo vision, one performs stereo calibration, the results of which are used for 3D reconstruction of points imaged in the two cameras. A common and flexible technique for such calibration is based on collection and processing pairs of images of a planar chessboard calibration pattern. The inherent weakness of this approach lies in its reliance on the random nature of data collection, which might lead to better or worse calibration results, depending on the collected set of image pairs. In this paper, a subset-based approach to camera and stereo calibration, along with its implementation based on OpenCV, is presented. It utilizes a series of calibration runs based on randomly chosen subsets from the global set of image pairs, with subsequent evaluation of metrics based on triangulating the features in each image pair. The proposed method is evaluated on a collected set of chessboard image pairs obtained with two identical industrial cameras. To highlight the capabilities of the method to select the best-performing calibration parameters, a principal component analysis and clustering of the transformed data was performed, based on the set of metric measurements per each calibration run.

## INTRODUCTION

Vision systems based on stereo imaging setups are widely used in numerous application contexts where contact-less measurement of 3D space is required. To perform 3D reconstruction, the imaging setup has to be calibrated. This includes calibration of each individual camera, combined with calibration of the whole stereo rig. In practice, a very widely used family of methods for camera and stereo calibration are based on a planar object with known pattern (such as a chessboard). The core principle of planar-based calibration is minimization of reprojection error in each image, as described in the original article by *Zhang (2000)*. Widely used implementation of this approach is available as a part of the open source computer vision library OpenCV.

Because stereo vision systems are pervasive in many optical measurement applications, the topics of stereo calibration and improvement of thereof have been widely presented in the research literature. Some approaches use the standard calibration approach directly. For example, the standard routine as implemented with OpenCV is applied by *Zhong & Dong (2015)* to guide a REINOVO robot using stereo vision. *Singh, Kumar & Nongmeikapam (2020)* combine the standard calibration approach with the preliminary

Corresponding author
Oleksandr Semeniuta,
oleksandr.semeniuta@ntnu.no

knowledge on the ranges of dimensions to be measured, namely height of a person. In other cases, some parts of the standard calibration pipeline are used in combination with additional supporting elements of the physical setup. Stereo calibration is tackled in context of a custom 3D scanner that includes a stereo camera rig (*Lin, 2020*). The solution makes use of extrinsic transformations with respect to the chessboard calibration object. Each camera is calibrated separately, with the relative transformation between the cameras computed from the known extrinsics. *Möller et al. (2016)* combine a stereo camera system with a laser tracker to increase absolute positional accuracy of a milling robot. The system relies on retroreflective markers on the robot tool to facilitate pose estimation. Finally, there exist approaches that greatly differ from the standard ones due to precise management of a custom geometric setup. An example is a custom calibration method based on the geometric constraints of the imaging setup, such as parallelism of the image planes and the calibration/measurement plane (*Ramírez-Hernández et al., 2020*).

When using the traditional planar calibration method, given enough images of the calibration target in different orientation, one can obtain sufficient calibration results. However, these results are slightly different depending on the actual input image set. Such property is known to planar-based calibration methods, and can be attributed to degenerate configuration due to many planes with similar orientation (*Zhang, 2000*). To make the calibration results better, a number of heuristics exist, such as the following: the view of the planar calibration target shall not be parallel in two or more images; for better estimation of the camera distortion, the calibration target shall appear in all four corners of the image and cover as much exterior orientations as possible; the best results would be obtained providing more than 10 calibration object views and orientation angle near 45° (*Zhang, 2000*; *Steger, Ulrich & Wiedemann, 2007*).

Since the plane-based calibration is usually performed manually by rotating a calibration target in front of one or more cameras, it may be tricky to precisely account for all the abovementioned heuristics. A sensible alternative can be to gather a relatively large set of images (or image pairs in the case of a stereo system) and evaluate how different subsets of the images work as an input to the standard calibration procedures. This approach has been investigated in a previous work by the author (*Semeniuta, 2016*), with an observation that the calibration parameters in a general form can be described by Gaussian distributions if numerous calibration runs with different subsets are invoked. However, it is challenging to select an optimal sample from such distribution if only one camera is used. It is worth noting that the original paper describing the plane-based method for calibrating a single camera (*Zhang, 2000*) also applied a similar approach, namely selecting subsets of four images out of five, when analyzing variability of calibration results.

The general approach in geometric computer vision is to formulate an estimation problem as a linear model, which can be solved in a closed form, and then apply iterative optimization to further refine the sought parameters given a nonlinear cost function that is closer to the reality (*Hartley & Zisserman, 2004*). Such cost function has to be differentiable to apply gradient-based optimization methods. In the case of camera and stereo calibration, this cost function models the reprojection error: given the known

real-world coordinates, how close their transformed values will be to the known coordinates of the corresponding image features? This cost function is well-suited for estimation of camera intrinsic parameters and relative pose of two cameras in a stereo system, however, it operates on a totally different domain than what one is interested during stereo reconstruction (image plane vs metric $\mathbb{R}^3$).

This paper proposes a subset-based stereo calibration method that performs a number of calibration runs on randomly selected subsets from the global set of image pairs and evaluates the quality of a calibration run by performing stereo reconstruction given all the image pairs and analyzing a cascade of metrics based on the reconstructed point clouds. The proposed approach does not aim to replace the well-known planar calibration method, but to conduct the latter in an automated manner according to the set of the proposed rules. As such, a more optimal calibration result can be obtained by automatic analysis of the set of individual calibration runs.

This paper applies the proposed method to a set of experimentally collected pairs of images acquired with a stereo rig. The analysis shows that in approximately half of the calibration runs, the performance is unsatisfactory. For the rest of the runs, one can rank them in the order of how well they perform, and hence, select the best-performing one. In addition to the analysis of the raw metric data obtained based on the triangulation results, principal component analysis (PCA) is performed on the vectors of metrics per each calibration run to transform the data into two components. PCA, along with k-means clustering, highlight the clear separation of the satisfactory and unsatisfactory classes and demonstrate the efficacy of the ranking.

The paper is organized as follows. First, a detailed overview of mathematical preliminaries of camera and stereo calibration is presented, followed with the specifics of OpenCV functionality and how it was harnessed in the presented work. Further, the proposed method is formulated, followed with its experimental validation and outlook for further research activities.

## CAMERA AND STEREO CALIBRATION

Geometric computer vision deals with estimating geometric properties of the scene being imaged, parameters of camera models, multiple-view geometry, and related aspects of computer vision (*Hartley & Zisserman, 2004*). Models for these tasks are normally expressed in linear form by operating with homogeneous vectors on the *projective space*. The latter constitutes an extension of the Euclidean space with points at infinity. Taking an example of Euclidean 2-space ($\mathbb{R}^2$), a point $(x, y)^T \in \mathbb{R}^2$ is represented in projective 2-space $\mathbb{P}^2$ as an equivalence class of coordinate triples $(kx, ky, k)^T$, where $k \neq 0$. Such triple is referred to as a *homogeneous vector*. A set of homogeneous vectors $\{(x, y, 0)^T\}$ correspond to points at infinity, each characterized by ratio $x : y$. A homogeneous vector $(0, 0, 0)^T$ is not an element of $\mathbb{P}^2$.

For camera calibration and 3D reconstruction, the pinhole camera model, the distortion model, and the epipolar geometry are used as a basis. The pinhole camera model deals with two coordinate frames (see Fig. 1): camera coordinate frame $\{C\}$ with axes $(x_c, y_c, z_c)$, located in the optical center, with $z_c$ perpendicular to the image plane, and the image

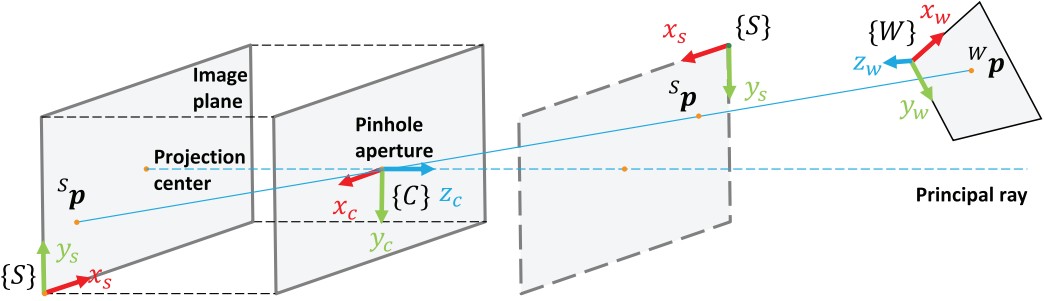

**Figure 1  Coordinate frames in the pinhole camera model.**

(sensor) coordinate frame $\{S\}$ with axes $(x_s, y_s)$, located in the top left corner of the image plane with $x_s$ and $y_s$ representing pixel columns and rows respectively.

The process of projection of points in the world coordinate frame $\{W\}$ to image pixels in $\{S\}$ using the pinhole model (expressed in the projective spaces $\mathbb{P}^3$ and $\mathbb{P}^2$ respectively) is referred to as a *perspective projection model*. The latter is a linear mapping $(x_w, y_w, z_w, 1)^T \mapsto (\lambda x_s, \lambda y_s, \lambda)^T$ that can be expressed in the matrix form as follows:

$$\begin{bmatrix} x_s \cdot \lambda \\ y_s \cdot \lambda \\ \lambda \end{bmatrix} = \begin{bmatrix} f_x & 0 & c_x \\ 0 & f_y & c_y \\ 0 & 0 & 1 \end{bmatrix} \begin{bmatrix} 1 & 0 & 0 & 0 \\ 0 & 1 & 0 & 0 \\ 0 & 0 & 1 & 0 \end{bmatrix} \begin{bmatrix} | & | & | & t_x \\ \mathbf{r_x} & \mathbf{r_y} & \mathbf{r_z} & t_y \\ | & | & | & t_z \\ 0 & 0 & 0 & 1 \end{bmatrix} \begin{bmatrix} x_w \\ y_w \\ z_w \\ 1 \end{bmatrix} \tag{1}$$

Or, for short:

$$\lambda \begin{bmatrix} x_s \\ y_s \\ 1 \end{bmatrix} = \mathbf{K}[\mathbf{I}_{3\times 3}|\mathbf{0}]^{\mathbf{c}}\mathbf{T_w} \begin{bmatrix} x_w \\ y_w \\ z_w \\ 1 \end{bmatrix} \tag{2}$$

where $\mathbf{K}$ is a camera matrix, which embodies such intrinsic parameters as the camera center $(c_x, x_y)$ and the focal lengths $(f_x, f_y)$, and $^{\mathbf{c}}\mathbf{T_w} \in SE(3)$ corresponds to a homogeneous transformation expressing coordinate frame $\{W\}$ in terms of coordinate frame $\{C\}$.

The distortion model $L : \mathbb{R}^2 \to \mathbb{R}^2$ describes a non-linear mapping from an "as-if perfectly imaged" image coordinate $(x, y)^T$ to the displaced image coordinate $(x^\star, y^\star)^T$ distorted by lens. One is normally interested in the inverse function $L^{-1}$ modeling the undistortion process, which is is characterized by radial distortion coefficients $(k_1, k_2, k_3)$ and tangential distortion coefficients $(p_1, p_2)$:

$$\begin{bmatrix} x \\ y \end{bmatrix} = L^{-1}(\begin{bmatrix} x^* \\ y^* \end{bmatrix}) = \begin{bmatrix} x^*(1 + k_1 r^2 + k_2 r^4 + k_3 r^6) + (2p_1 y + p_2(r^2 + 2x^2)) \\ y^*(1 + k_1 r^2 + k_2 r^4 + k_3 r^6) + (p_1(r^2 + 2y^2) + 2p_1 x) \end{bmatrix} \tag{3}$$

where r is the radial distance of point $(x^\star, y^\star)^T$.

Let $Cam_{\mathbf{p},\xi} : \mathbb{P}^3 \to \mathbb{R}^2$ be a full camera model that embodies both perspective projection and non-linear distortion effect. $Cam_{\mathbf{p}, \xi}$ is parameterized by intrinsic parameters $\mathbf{p} = (f_x, f_y, c_x, c_y, k_1, k_2, k_3, p_1, p_2)^T$ and extrinsic parameters describing the pose $\xi$ of $\{W\}$ with

respect to $\{C\}$, typically described with a homogeneous transformation $^{c}\mathbf{T_w}$. $Cam_{\mathbf{p},\xi}$ first performs linear transformation described in Eq. (2), followed by lens-induced distortion of the projected point $(x_s, y_s)$:

$$\begin{bmatrix} x_s^* \\ y_s^* \end{bmatrix} = L\left(\begin{bmatrix} x_s \\ y_s \end{bmatrix}\right) \tag{4}$$

The goal of camera calibration is to estimate camera intrinsic parameters—$(f_x, f_y, c_x, c_y)$ embodied in the camera matrix $\mathbf{K}$ (2), as well as parameters $(k_1, k_2, k_3, p_1, p_2)$ of the non-linear distortion model (3).

The general principle of camera calibration lies in finding the correspondence between a sufficiently large number of known 3D points and their projections in the image (*Steger, Ulrich & Wiedemann, 2007*). The known points are provided by the calibration object containing features that have known coordinates and are easily identifiable by image processing algorithms. As described by *Zhang (2004)*, a calibration object may be different depending on one of the respective calibration techniques: (1) a precisely manufactured 3D object (typically, consisting of three perpendicular planes) (*Tsai, 1987*; *Heikkila, 2000*), (2) a planar object (*Zhang, 2000*; *Sturm & Maybank, 1999*), or (3) three or more collinear points (string of balls) (*Zhang, 2004*). Because manufacturing of a custom 3D object is costly, planar objects make calibration process more flexible, and are widely used in practice.

The camera takes $k$ images of the planar calibration object from different views. For each view $i$, a homography matrix $\mathbf{H_i}$ is computed based on two sets of points: (1) real-world coordinates of the target points in the world coordinate frame, and (2) their projected pixel values determined with the appropriate feature detection technique. A system of homogeneous equations $\mathbf{Ax} = 0$ is formed from the homography matrices $\{\mathbf{H_i}\}$ and the constraints imposed by the orthonormality of the rotation vectors, and solved in closed form to determine the initial values for the the parameters embodied in the camera matrix $\mathbf{K}$. The latter is derived from $\mathbf{x}$, the extrinsic parameters for each view $^{c}\mathbf{T_{wi}}$ are computed given $\mathbf{H_i}$ and $\mathbf{K}$.

Given the initial values of $\mathbf{K}$ and the set of poses for each view $\{^{c}\mathbf{T_{wi}}\}$, the values of all intrinsic parameters $\mathbf{p} = (f_x, f_y, c_x, c_y, k_1, k_2, k_3, p_1, p_2)^T$ are refined by non-linear estimation minimizing the reprojection error.

The reprojection error is defined as follows. Let $m$ be the number of features used for calibration. Set $W = \{\mathbf{x_i} \in \mathbb{P}^3, i = 1, \ldots m\}$ contains known coordinates of the points in the coordinate frame of the calibration object. Set $S = \{^{S}\mathbf{x_i} \in \mathbb{R}^2, i = 1, \ldots m\}$ contains the corresponding coordinates of image features. $^{S}\mathbf{x_i}$, has 2 degrees of freedom, which are the column and the row coordinate of a point in the image: $^{S}\mathbf{x_i} = (x_i, y_i)^T$. A known 3D point $\mathbf{x_i}$ is reprojected to the image as follows:

$$\begin{bmatrix} x_i^{(rp)} \\ y_i^{(rp)} \end{bmatrix} = Cam_{\mathbf{p},\xi_i}(\mathbf{x_i}) \tag{5}$$

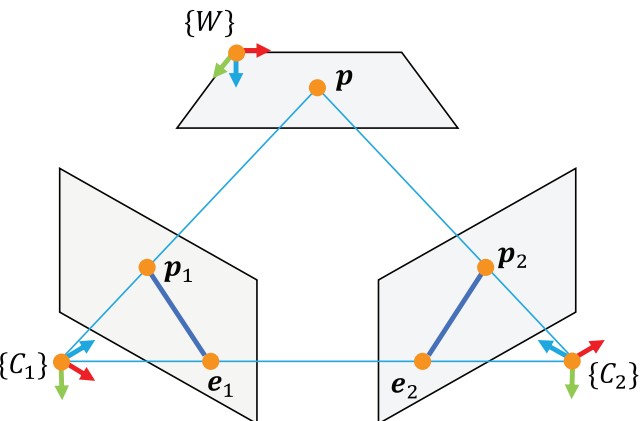

**Figure 2 Epipolar geometry.**   

The misfit function $\mathbf{f}_{rp}$ maps $\mathbf{p}$ to $\mathbb{R}^{2m}$, as for each feature point, two difference values are obtained:

$$\mathbf{f}_{rp}(\mathbf{p}) = \begin{bmatrix} \begin{bmatrix} x_1 \\ y_1 \end{bmatrix} - Cam_{\mathbf{p},\xi_1}(\mathbf{x_1}) \\ \ldots \\ \begin{bmatrix} x_m \\ y_m \end{bmatrix} - Cam_{\mathbf{p},\xi_m}(\mathbf{x_m}) \end{bmatrix} \tag{6}$$

Given $\mathbf{f}_{rp}$, the reprojection error constitutes a root mean square cost function:

$$C_{rp}(\mathbf{p}) = \sqrt{\frac{1}{2m}\mathbf{f}_{rp}(\mathbf{p})^T \mathbf{f}_{rp}(\mathbf{p})} \tag{7}$$

In the OpenCV implementation, distortion coefficients are computed using the method proposed by *Brown (1971)*, and, based on them, $\mathbf{K}$ is reestimated (*Bradski & Kaehler, 2008*).

Calibration of stereo vision system is based on epipolar geometry (see Fig. 2), where a 3D point is imaged in two cameras, with the corresponding projections obeying a set of geometric constraints. The latter can be expressed in term of essential matrix $\mathbf{E}$ and fundamental matrix $\mathbf{F}$. The fundamental matrix constrains two corresponding points in the image coordinates $\mathbf{x}_1$ and $\mathbf{x}_2$ by the following relation:

$$\mathbf{x}_1 \mathbf{F} \mathbf{x}_2 = 0 \tag{8}$$

A similar relation holds for points expressed in normalized coordinates $\tilde{\mathbf{x}}_1$ and $\tilde{\mathbf{x}}_2$, i.e. assuming camera matrices $\mathbf{K}_1$ and $\mathbf{K}_2$ are unit matrices:

$$\tilde{\mathbf{x}}_1 \mathbf{E} \tilde{\mathbf{x}}_2 = 0 \tag{9}$$

The essential matrix encodes the relative rotation $\Omega$ and translation $\mathbf{t}$ between the cameras:

$$\mathbf{E} = skew(\mathbf{t})\Omega = \begin{bmatrix} 0 & -t_z & t_y \\ t_z & 0 & -t_x \\ -t_y & t_x & 0 \end{bmatrix} \begin{bmatrix} | & | & | \\ \mathbf{r_x} & \mathbf{r_y} & \mathbf{r_z} \\ | & | & | \end{bmatrix} \tag{10}$$

Hence, the rotation and translation can be decomposed from $\mathbf{E}$ once it is estimated from the point correspondences. In the complete stereo calibration pipeline, the fundamental matrix $\mathbf{F}$ is first estimated given the original image points. Further, the essential matrix is obtained given the previously estimated camera matrices:

$$\mathbf{E} = \mathbf{K}_2^T \mathbf{F} \mathbf{K}_1 \tag{11}$$

The relative rotation $\Omega$ and translation $\mathbf{t}$ of the second camera expressed in terms of the first one, decomposed from $\mathbf{E}$, are used as a basis, together with the intrinsic parameters of the cameras, to triangulate pairs of corresponding image points.

## IMPLEMENTATION SPECIFICS

All the work underlying this paper has been implemented on top of the basic calibration routines in the OpenCV library using the Python API. As such, the implementation specifics presented in this section, refer to specific OpenCV functions as defined in the API. The goal of this section to present the workflow of the chessboard corners preparation, camera and stereo calibration, and stereo triangulation in as transparent way as possible.

In OpenCV, the stereo calibration pipeline starts with the camera calibration process for the two cameras, followed by the stereo calibration process, the results of which are fed to the computation of rectification transform. The latter step is important, since both dense and sparse stereo require that the input image points are undistorted and rectified.

The core parameters estimated during camera and stereo calibration are the following:

1. Camera matrix $\mathbf{K}_1$ and a vector of distortion coefficients $\mathbf{d}_1$ for the first camera.
2. Camera matrix $\mathbf{K}_2$ and a vector of distortion coefficients $\mathbf{d}_2$ for the second camera.
3. Relative rotation $\Omega$ and translation $\mathbf{t}$ of the second camera expressed in terms of the first camera.
4. Rotation matrices $\mathbf{R}_1$ and $\mathbf{R}_2$ describing the rectification transforms of the two cameras.
5. Projection matrices $\mathbf{P}_1$ and $\mathbf{P}_2$ in the rectified coordinate systems.

All the listed key parameters are used to perform undistortion and rectification of two sets of image points. After this, $\mathbf{P}_1$ and $\mathbf{P}_2$ are used to perform triangulation.

Below, the logic of corners preparation and stereo calibration is modeled in terms of EPypes (*Semeniuta & Falkman, 2019*) directed acyclic graphs. The image processing logic is arranged as an executable direct acyclic bipartite graph $G$:

$$G = (F, T, E)$$

where $F$ is a set of functions, $T$ is a set of data tokens, and $E$ is a set of directed edges between functions and tokens and vice-versa. A function $f \in F$ is associated with a Python callable. A token $t \in T$ represents a data object of an arbitrary type. The tokens highlighted in gray represent fixed values that parameterize the algorithms. The source tokens that are not fixed represent inputs to the algorithm.

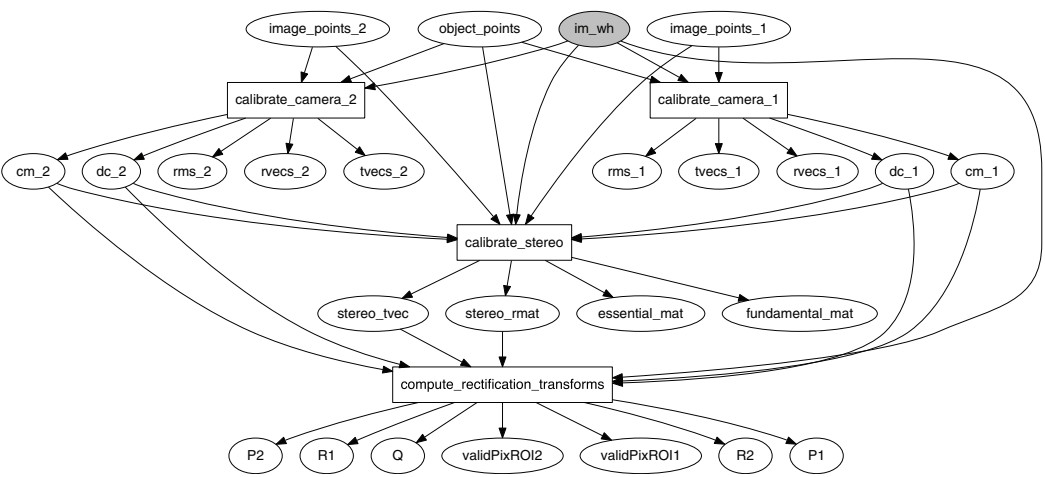

**Figure 3 Computational graph for the process of stereo calibration.**

The graph in Fig. 3 models the stereo calibration process. Functions `calibrate_camera_1` and `calibrate_camera_2` encapsulate the `cv2.calibrateCamera` function, performing calibration of each individual camera in the stereo rig. Similarly, `calibrate_stereo` encapsulates `cv2.stereoCalibrate`, estimating the transformation between the two cameras, and `compute_rectification_transforms` encapsulates `cv2.stereoRectify`, computing the transformations that undistorts and align a pair of images so that a feature detected by the first camera appears in the second image in the same row.

The graph in Fig. 4 processes the collected set of image pairs by attempting to find chessboard corners in both images. Here `calibration_images_1` and `calibration_images_2` represent two identically-sized lists of images from the first and the second camera respectively. If $N$ is the total number of image pairs, the pairs are indexed as integers in the set $P = \{0, 1, 2,\ldots, N-1\}$. Indices of the pairs where the corners were *successfully identified in both images* ($P_{both} \subseteq P$) are retained in the `indices` token.

Triangulation has to be preceded with undistortion of image points. To perform undistortion (using `cv2.undistortPoints`) of points detected in an image from camera $i \in \{1, 2\}$, camera matrix $\mathbf{K}_i$, distortion coefficients $\mathbf{d}_i$, and matrices $\mathbf{P}_i$, and $\mathbf{R}_i$ are used.

## PROPOSED METHOD

For the total set of all available image pairs, identify corners in them and store the corners for further processing. Additionally, store indices of the image pairs that resulted in successful corners detection in both images (see Fig. 4).

Given a set of all image pairs, randomly select a series of $m$ integers $S = \{s_1,\ldots, s_m\}$, within the predefined range $[s_{min}, s_{max}]$. The value of $s_i$ corresponds to the size of image subset used for calibration. For each $i \in \{1,\ldots, m\}$, randomly select $s_i$ image pairs and perform stereo calibration (here it is presumed that all image pairs have been validated to retain only those with successfully identified corners in both images, totaling to $n$ pairs). Store the result of each calibration run $i$.

**Peer**J Computer Science

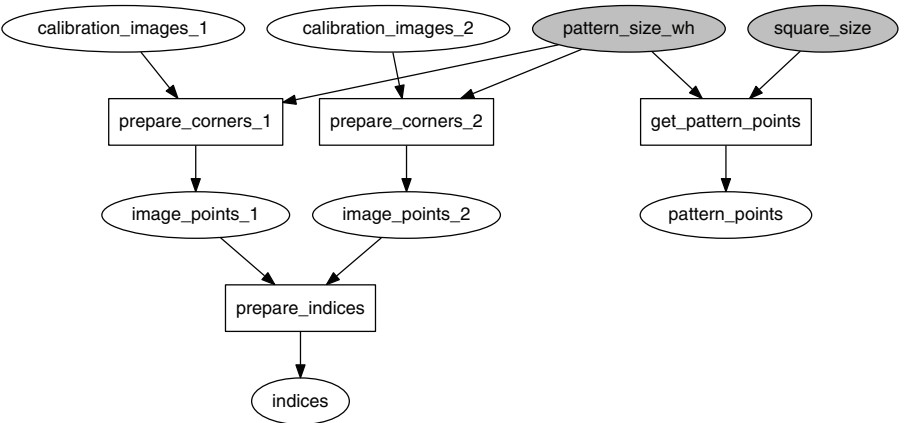

**Figure 4 Computational graph for the process of image points preparations.**

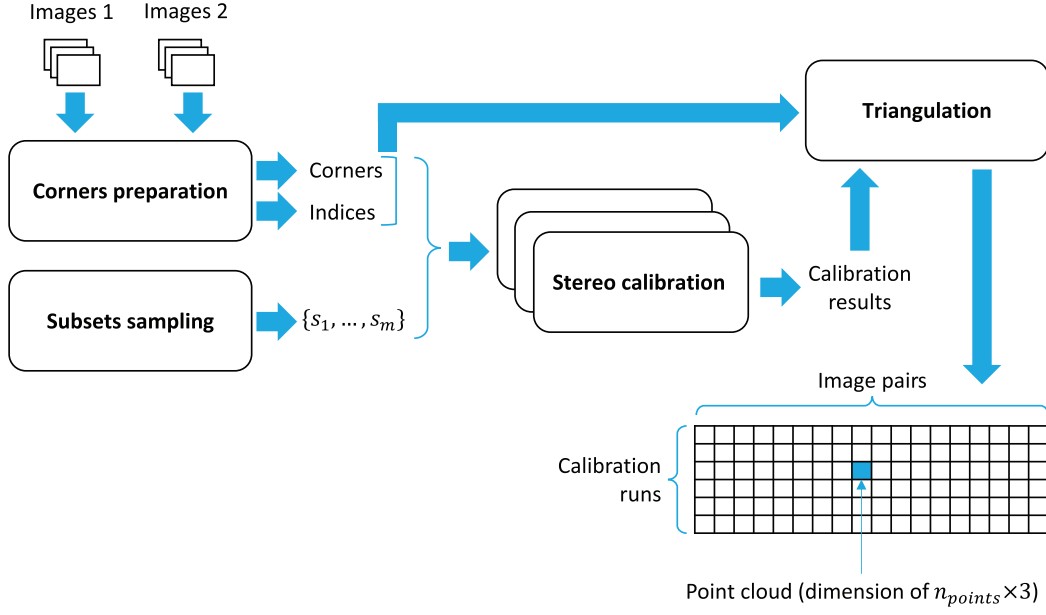

**Figure 5 Workflow of point clouds' preparation.**

The general idea of the proposed method is to *utilize the set of all image pairs* that capture the calibration object in different orientation as a basis for assessment of stereo triangulation quality. Let $i \in \{1,\ldots,m\}$ denote the index of a calibration set, and let $j \in \{1,\ldots,n\}$ denote the index of an image pair. There exists $mn$ such combinations of calibration results and image pairs.

Given calibration run $i$ and image pair $j$, triangulate the coordinates of the corner points. As a result of triangulation, a point cloud in $\mathbb{R}^3$ is obtained. All the point clouds are stored in a tensor of shape ($m \times n \times n_{points} \times 3$). The overall workflow of point clouds' preparation is presented in Fig. 5.

In this work, the point cloud represents the chessboard corners, and hence, the triangulated points have to be as close to the original chessboard geometry as possible.

This includes the distances between the neighboring points and how much the triangulated points resemble a plane. Given a point cloud as $n_{points}$ points in $\mathbb{R}^3$, a *point cloud metric function* maps these point to a single value in $\mathbb{R}$. Given the properties imposed by the chessboard geometry, namely coplanarity and equal distances between the neighboring points, two such metrics are defined:

1. Function $J_{MDIR}$ measures the mean distances of neighboring points in rows. It should ideally be close to the nominal square size.
2. Function $J_{PDRMS}$ measures plane difference RMS error: given the points in $\mathbb{R}^3$, fit a plane to them, and calculate the RMS error of the points with respect to the plane. In the ideal case, the overall error should be close to zero, showing the evidence of a good plane reconstruction.

A calibration run $i \in \{1,\ldots, m\}$ is associated with $n$ point clouds obtained with triangulation of the respective image pairs. For each of these point clouds, the metric functions $J_{MDIR}$ and $J_{PDRMS}$ are computed. Let $d_{ij}$ denote the value of $J_{MDIR}$ for image pair $j$ given calibration $i$ and $d$ denote the nominal value of the square size. Given a tolerance value $\delta \in \mathbb{R}^+$, the following relation indicates an acceptable triangulation result, namely the one that is not an outlier:

$$|d_{ij} - d| < \delta \tag{12}$$

Let $p_{ij}$ denote the value of $J_{PDRMS}$ for image pair $j$ given calibration $i$ and $p_{max}$ denote the maximal value of $J_{PDRMS}$ that separates inliers and outliers.

For each combination of a set of calibration parameters ($i$) and image pair ($j$), an indicator variable $g_{ij} \in \{0, 1\}$ is defined as follows:

$$g_{ij} = \begin{pmatrix} 1, & if\,(|d_{ij} - d| < \delta) \wedge (p_{ij} < p_{max}) \\ 0, & \text{otherwise.} \end{pmatrix} \tag{13}$$

The number of acceptable triangulation results $a_i$ per calibration run is thus defined as follows:

$$a_i = \sum_{j=1}^{n} g_{ij} \tag{14}$$

Based of the values of $g_{ij}$, which serve a role of a mask for the cases with acceptable triangulation, vectors $\mathbf{d}_i$ and $\mathbf{p}_i$ are defined. They contain the values of the triangulation metrics $J_{MDIR}$ and $J_{PDRMS}$ respectively only for those $(i, j)$ that are characterized by $g_{ij} = 1$.

To assess the calibration performance based on triangulation, a number of metrics is proposed (for a calibration run $i$), which are defined below.

1. The number of image pairs that resulted in an acceptable triangulation result, namely $a_i$.
2. Mean and standard deviation of the $J_{MDIR}$ scalar triangulation assessment metric for such $(i, j)$ that are masked by the condition of $g_{ij} = 1$:

$$\mu_i = \bar{\mathbf{d}}_i = \frac{1}{a_i} \sum_{j=1}^{n} (d_{ij} \text{ if } g_{ij} = 1) \tag{15}$$

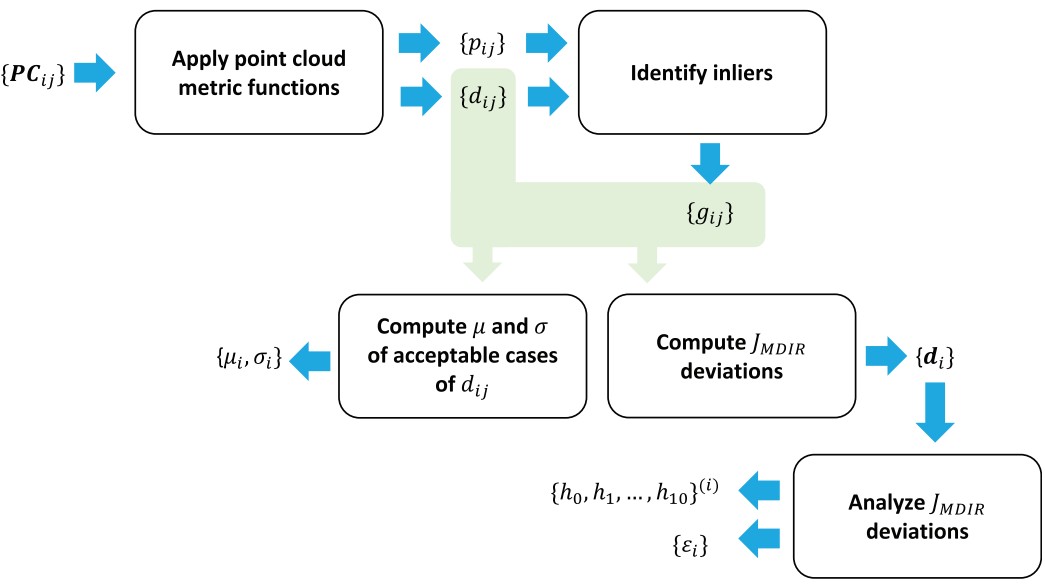

**Figure 6** Metrics computation workflow.               

$$\sigma_i = \sqrt{\frac{1}{a_i - 1} \sum_{j=1}^{n} ((d_{ij} - \bar{\mathbf{d}}_i)^2 \text{ if } g_{ij} = 1)} \tag{16}$$

3. Number of image pairs $h_0^{(i)}$ that fell into the first histogram bin (the one closest to zero) of the *distribution of errors from the nominal scalar triangulation assessment metric*. The histogram is constructed as follows. A vector $\mathbf{e}$ of absolute errors is computed given the vector $\mathbf{d}_i$:

$$\mathbf{e}_i = |\mathbf{d}_i - d| \tag{17}$$

A value $\Delta \in \mathbb{R}^+$ is defined, and the range $[0, \Delta]$ is split into 10 bins, with the first bin corresponding to the range of $[0, \Delta/10)$. The values in $\mathbf{e}_i$ are distributed across the defined 10 bins, with those $\varepsilon_{ij} \geq \delta$ falling into the eleventh bin. As such, if for example $\Delta = 1$ mm, the number of image pairs that fall into the first bin indicate how many of those resulted in the value of $0 \leq |d_{ij} - d| \leq 0.1$ mm. If this number is bigger for a calibration $i$, it is indicative of a more optimal triangulation.

4. Maximal absolute error from the nominal scalar triangulation assessment metric:

$$\varepsilon_i = max(\mathbf{e}) \tag{18}$$

Furthermore, all calibration runs are ranked from the best-performing to the worst-performing based on the associated values of the metrics defined above. The ranking logic proposed in this paper is as follows: sort the calibration runs first by the values of $h_0^{(i)}$ (in decreasing order) and then by the values of $\varepsilon_i$ (in increasing order). Then, pick the calibration results associated with the calibration run of rank 1 as the final results.

The overall workflow of metrics computation is presented in Fig. 6.

The proposed method has been implemented as an open-source Python library, dubbed `vcalib`, based on OpenCV[1].

[1] The source code for the library is available at https://github.com/semeniuta/vcalib under the 3-clause BSD license.

**Table 1 Dimensions of the images and the calibration pattern.**

| Parameter | Value |
|---|---|
| Image dimensions | 1,360 × 1,024 pixels |
| Pattern size | 9 × 7 |
| Square size | 20 mm |

## EXPERIMENTAL VALIDATION

### Experimental setup and data collection

A stereo rig of two identical Prosilica GC1350 cameras with TAMRON 25-HB/12 lens is used to collect pairs of images with a chessboard calibration pattern. Allied Vision Vimba is used as an image acquisition driver, orchestrated by FxIS (*Semeniuta & Falkman, 2018*) to achieve synchronization across the two cameras.

The data collection component continuously receive streams of images from the two cameras. At certain moments, a pair of images is obtained that most closely correspond with their acquisition timestamps. They further undergo chessboard corner detection. In case the corners were successfully identified in both images, the latter are saved to disk.

In total, $n = 264$ image pairs are collected. Their dimensions, along with the details on the chessboard calibration object, are presented in Table 1.

### Experimental results

Given the collected set of $n = 264$ image pairs and the method described above, $m = 200$ calibration runs were invoked with the size of the randomly selected input subset varying between $s_{min} = 15$ and $s_{max} = 30$ image pairs. The resulting calibration parameters were used for triangulation using all images, and the results were collected in a tensor of dimension $200 \times 264 \times 63 \times 3$. The dimension of the latter two components correspond to the number of chessboard points in $\mathbb{R}^3$ ($n_{points} = 9 \times 7 = 63$).

Given the triangulation results, outliers were excluded. An outlier is defined as a calibration run/image pair combination for which either of the two following conditions holds true: $|J_{MDIR} - d| \geq 5$ mm (where $d$ is the nominal square size dimension), $J_{PDRMS} \geq 10$ mm (see the definition in Eq. (13)). The result of marking inliers and outliers is shown as a binary mask in Fig. 7. One may observe that certain calibration runs perform consistently well on the most of the image pairs. Similarly, some image pairs result in better triangulation results than the others.

The distributions of the two metric functions operating on the triangulated point clouds are shown in Fig. 8 ($J_{MDIR}$) and Fig. 9 ($J_{PDRMS}$). The blue vertical lines visualize the ranges of values that are considered as inliers according to Eq. (13). The green vertical lines denote the nominal values of the two metrics. Particularly for the $J_{MDIR}$ metric it can be seen that the triangulation in most of the cases is rather good, with most of the values tightly centered around the nominal dimension of $d = 20$ mm. When it comes to the flatness-related metric of $J_{PDRMS}$, most of the values are close to zero, but there is a substantial number of cases with higher values of the plane fit RMS error.

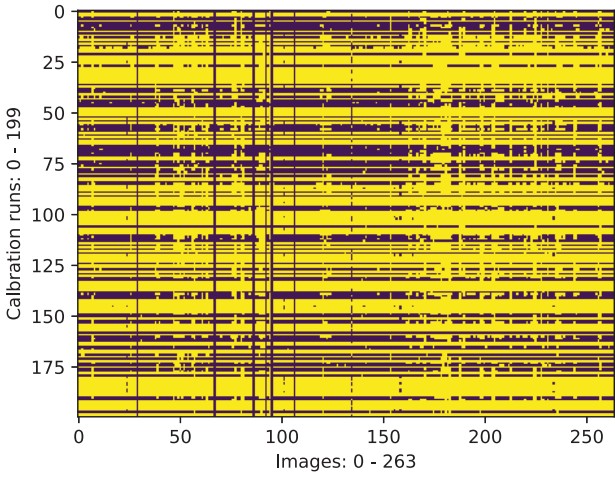

**Figure 7** Calibration run/image pair combinations that are characterized with acceptable values of the scalar triangulation metrics (yellow). 

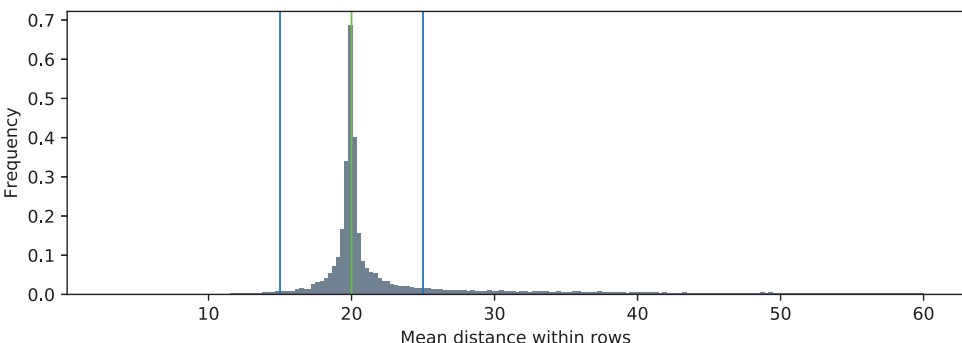

**Figure 8** Distribution of mean distance between neighboring points in rows ($J_{MDIR}$) and its relation to the nominal value (20 mm, green). The blue lines indicate the inlier range.
 

Based on the masked combinations of calibration runs and image pairs (as shown in Fig. 7) and the values of $J_{MDIR}$ and $J_{PDRMS}$, a set of metrics is computed for each calibration run: mean and standard deviation of the $J_{MDIR}$ values, $\mu_i$, $\sigma_i$, number of acceptable image pairs per calibration run, $a_i$, values of the histogram bins based on the absolute differences of $J_{MDIR}$ with the nominal value, $\{h_0, h_1, \ldots, h_{10}\}^{(i)}$, number of image pairs used for stereo calibration, $s_i$, and the mean value of $J_{PDMRS}$, $p_i$. Further in this section, the index $i$ is omitted when referring about the metrics.

The calibration runs are further ranked first by the values of $h_0$ (in decreasing order) and then by the values of $\varepsilon$ (in increasing order). The values of all metrics for the 10 best calibration runs are shown in Table 2. The first row of the table correspond to the best-performing calibration run (rank 1).

To better motivate the rankings of the calibration runs as shown in Table 2, it is worth to investigate the values of the histogram bins. The first histogram bin $h_0$ describes the

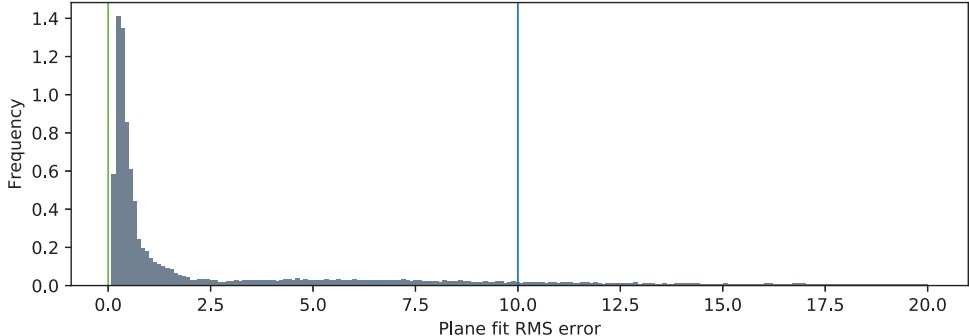

**Figure 9 Distribution of the plane fit RMS error ($J_{PDRMS}$). The green line represents the ideal value of the error ($p = 0$), and the blue line represents the value of $p_{max}$.**

**Table 2 Values of the calibration run metrics for five best calibration runs.**

| $i$ | $\mu$ | $\sigma$ | $a$ | $\varepsilon$ | $h_0$ | $h_1$ | $h_2$ | $h_3$ | $h_4$ | $h_5$ | $h_6$ | $h_7$ | $h_8$ | $h_9$ | $h_{10}$ | $s$ | $p$ |
|---|---|---|---|---|---|---|---|---|---|---|---|---|---|---|---|---|---|
| 195 | 20.090 | 0.951 | 257 | 4.310 | 158 | 25 | 5 | 6 | 4 | 4 | 4 | 3 | 1 | 4 | 43 | 24 | 0.495 |
| 92 | 20.074 | 0.969 | 257 | 4.492 | 142 | 19 | 17 | 13 | 5 | 4 | 6 | 2 | 1 | 5 | 43 | 25 | 0.501 |
| 154 | 20.066 | 0.948 | 257 | 4.206 | 140 | 23 | 22 | 9 | 4 | 6 | 2 | 2 | 1 | 3 | 45 | 16 | 0.541 |
| 193 | 20.157 | 0.986 | 257 | 4.366 | 134 | 22 | 20 | 14 | 9 | 4 | 2 | 6 | 2 | 2 | 42 | 17 | 0.578 |
| 103 | 20.162 | 0.981 | 257 | 4.671 | 133 | 47 | 7 | 7 | 4 | 4 | 4 | 0 | 2 | 4 | 45 | 27 | 0.509 |
| 163 | 20.064 | 0.962 | 257 | 4.335 | 131 | 24 | 20 | 9 | 12 | 4 | 5 | 3 | 1 | 3 | 45 | 27 | 0.492 |
| 159 | 20.061 | 0.934 | 257 | 4.061 | 128 | 36 | 20 | 8 | 8 | 3 | 4 | 2 | 3 | 2 | 43 | 23 | 0.501 |
| 122 | 20.075 | 0.964 | 257 | 4.369 | 124 | 45 | 14 | 8 | 5 | 5 | 6 | 1 | 1 | 5 | 43 | 15 | 0.516 |
| 51 | 20.044 | 0.926 | 257 | 4.143 | 120 | 32 | 25 | 10 | 13 | 2 | 4 | 3 | 3 | 3 | 42 | 17 | 0.504 |
| 33 | 20.046 | 0.976 | 258 | 4.999 | 112 | 34 | 16 | 17 | 13 | 10 | 4 | 2 | 2 | 4 | 44 | 17 | 0.540 |

number of image pairs in which the mean value of $\mathbf{e}_i$ is closest to zero. The last histogram bin $h_{10}$ describes the number of image pairs in which the mean value of $\mathbf{e}_i$ is beyond the specified top threshold $\Delta$. In the presented analysis, $\Delta = 1$ mm, which corresponds to the range of $[0, 0.1)$ mm for $h_0$ and $[1\ mm, \infty)$ for $h_{10}$. Figure 10 shows the histograms for the calibration runs of rank 1 (the best), rank 25, rank 50, rank 75, rank 100, and rank 125. It is clear that the selected ranking clearly sorts the alternative calibration runs from the best performing to the worst performing one.

Given the full dataset of metrics per calibration runs, a scatter plot of $h_0$ against $\varepsilon$ is visualized in Fig. 11. The color of each dot represents the value of $a$ for this calibration run. It can be seen that there are clearly two classes of calibration runs, namely those with acceptable calibration results and those with non-satisfactory results, obtained due to degenerate configuration of the input data sets.

## Experiment assessment

To highlight the details of the conducted experiment, two visualizations are provided in this section. First, the coverage of the two image planes by chessboard points are shown in

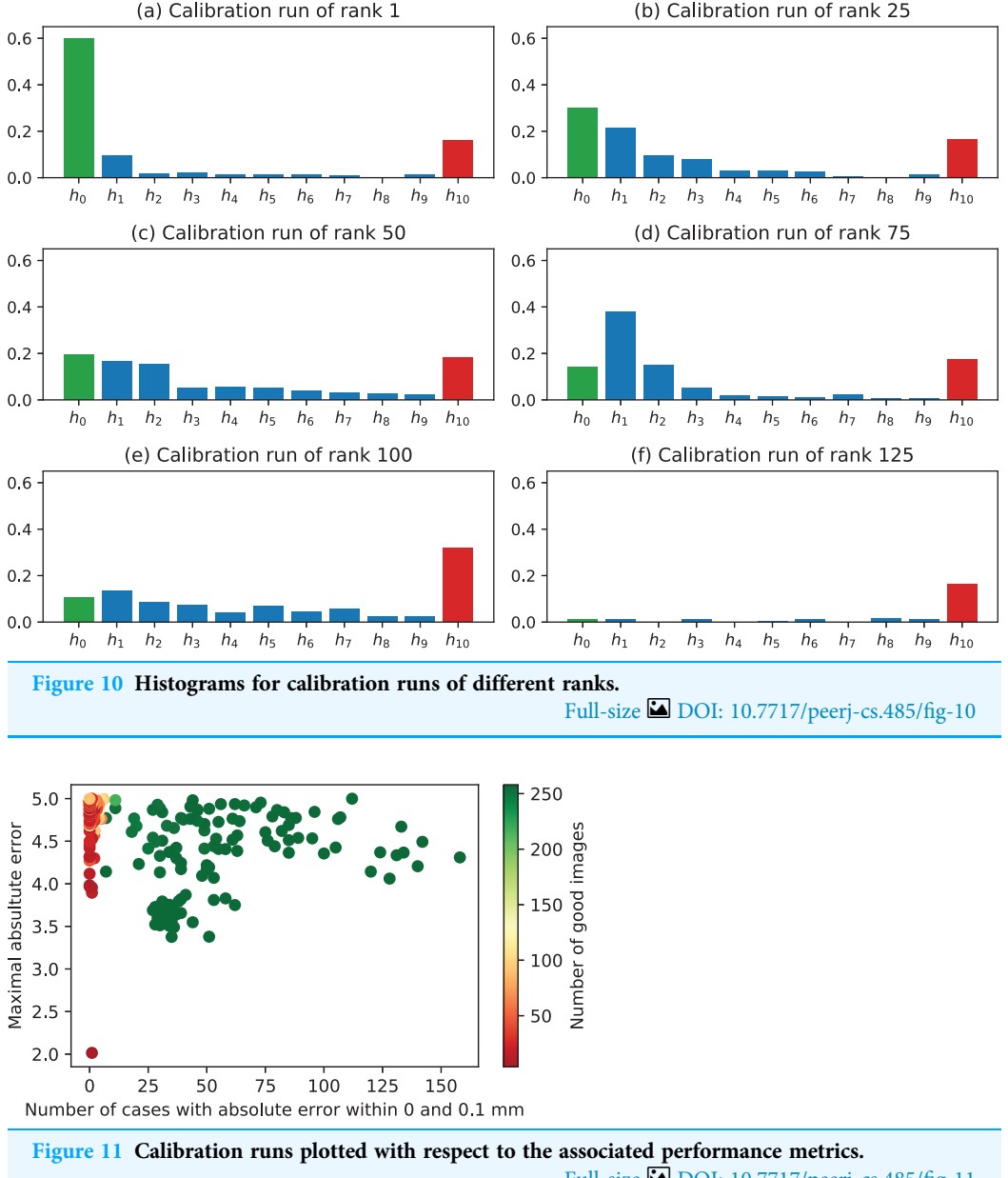

**Figure 10 Histograms for calibration runs of different ranks.**

**Figure 11 Calibration runs plotted with respect to the associated performance metrics.**

Fig. 12. The points plotted in yellow constitute the total set of all chessboard corners. The points plotted in blue correspond to the subset of those contained in the image pairs forming the highest-ranked calibration run.

It can be seen that, overall, most of the respective image planes are covered with points. Because the images were acquired so that the whole calibration plane is fully visible in both cameras, the left side in the left image and the right side in the right image are not covered with points.

Additionally, Fig. 13 shows the distribution of $z$-translation fo the pose of the calibration object with respect to the left camera. These measurements were conducted based on performing pose estimation (`cv2.solvePnP` function) of all chessboard views in the left

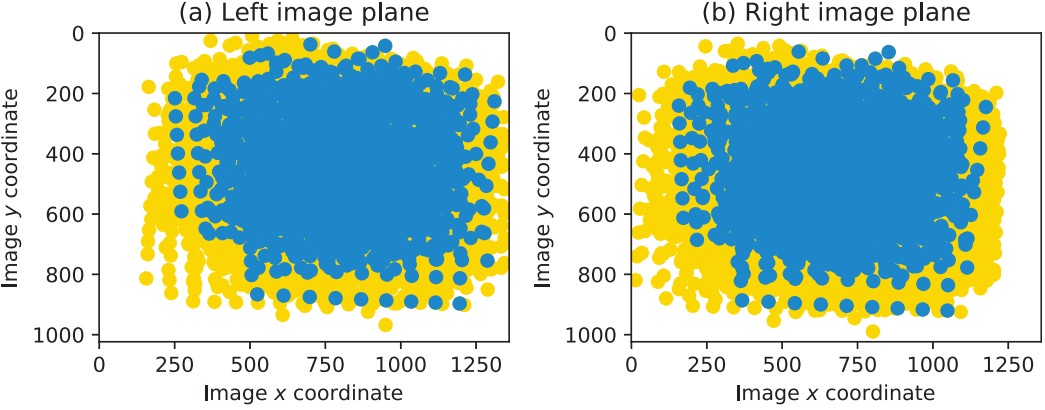

**Figure 12 Coverage of the left and the right image planes by chessboard points.**

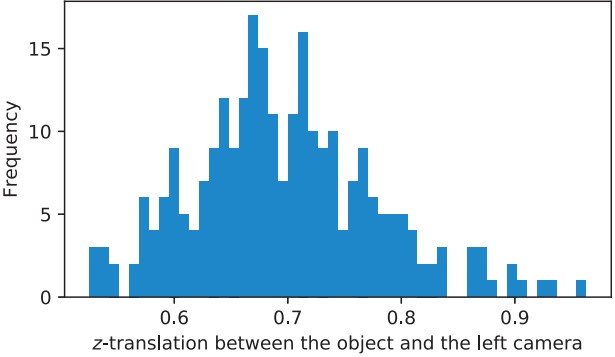

**Figure 13 Distribution of $z$-translation fo the pose of the calibration object with respect to the left camera.**

camera image given the highest-ranked calibration run. It can be seen that the operating range of the studied stereo rig is between 0.5 and 1 m.

## Accuracy assessment

To reason about accuracy of the proposed method, the following approach is applied. A single image pair is chosen as the benchmark. A suitable candidate for this is the image pair that results in acceptable triangulation results based on most calibration runs. Further, parameters from a number of first $n_{first}$ ranked calibration runs are used to triangulate the points from the chosen image pair. Distance between each neighboring points in rows ($J_{MDIR}$) are measured and collected in an array. Let this array be denoted as **a**.

For each calibration run, the deviation of the mean of the array **a** from the nominal value ($d = 20$ mm) is computed, along with the standard deviation of the values of **a**. Both of these metrics are desirable to be close to zero. When they are plotted together (in Fig. 14 for $n_{first} = 100$), with the calibration runs of ranks 1–20 and 21–40 marked with distinct colors, it may be observed, that on average, the calibration runs of higher ranks (closer to 1) result in triangulation of higher accuracy.

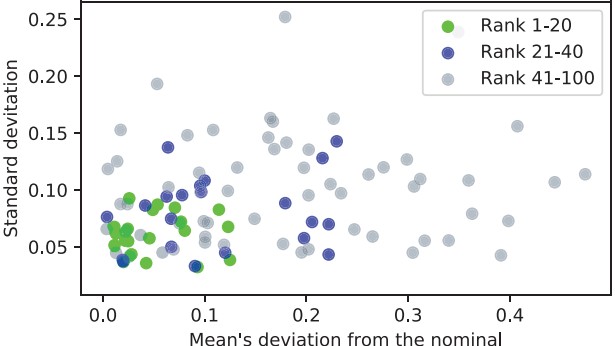

**Figure 14 Accuracy assessment of the first 100 ranked calibration runs.**

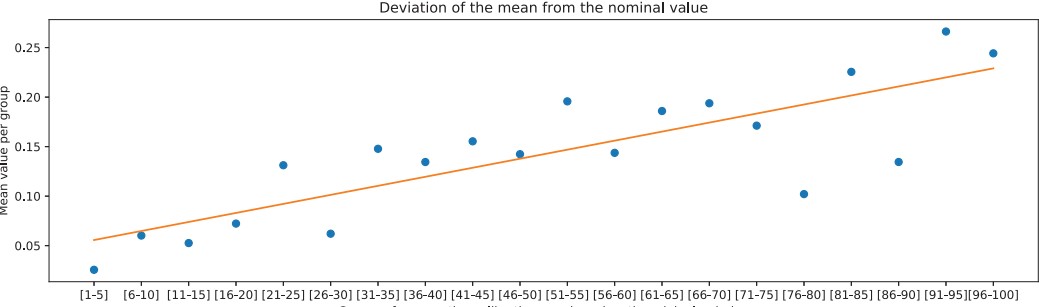

**Figure 15 Mean value of the deviation of $J_{MDIR}$ from the nominal value of the chessboard square of 20 mm, calculated for the groups of consecutive calibration runs, and the associated linear trend.**

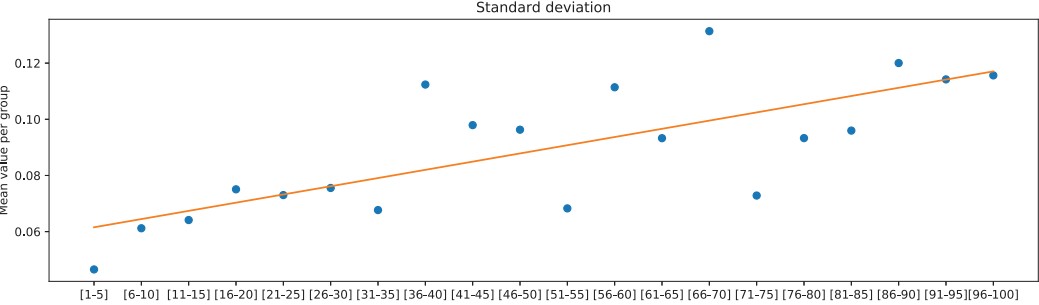

**Figure 16 Means of standard deviations of the measurements of distances between neighboring points in rows, calculated for the groups of consecutive calibration runs, and the associated linear trend.**

The overall trend can be further validated by comparing the two deviation metrics against groups of 5 consecutive ranked calibration runs. It can be observed in Figs. 15 and 16 that the deviations on average increase with decreasing the rank of the calibration runs (moving away from rank 1).

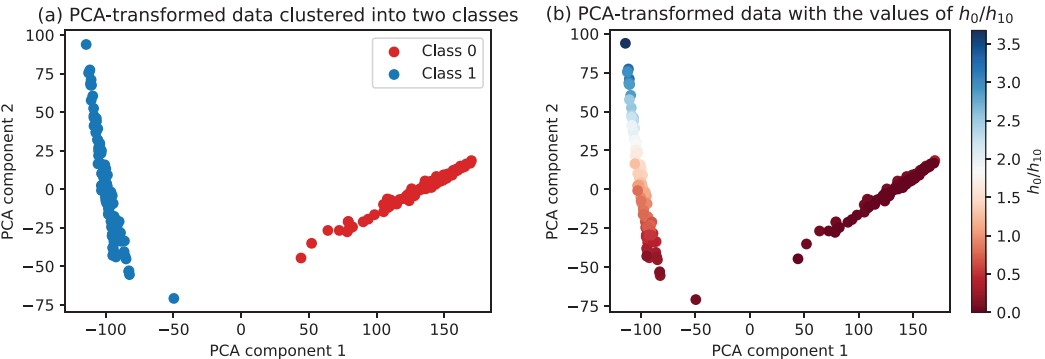

**Figure 17 Data transformed into two principal components.**

## Principal component analysis

To take into account all the metrics per calibration run, principal component analysis is performed. It takes the dataset as shown in Table 2 and maps the 17 features to the two dimensional plane that preserves most of the original variability in data. As such, this step integrates all the measurements yet allowing to visualize the dataset in two dimensions. As it can be seen in Fig. 17A, the PCA-transformed data clearly forms two distinct clusters. The green/red labeling in Fig. 17A has been automatically performed using k-means clustering with the PCA-transformed data as an input. The same data colored by the corresponding values of the $h_0/h_{10}$ ratio is presented in Fig. 17B. One can observe that class 0 (corresponding to the degenerate configurations) is associated with uniformly low values of the $h_0/h_{10}$ ratio, while class 1 (configurations with acceptable calibration results) vary in the range of the defined ratio, with some of them clearly better than the others.

## CONCLUSION AND FURTHER WORK

This paper has presented an approach for calibration of stereo vision setup that utilizes the standard technique for plane-based calibration, along with its OpenCV-based implementation, but with a novel technique for systematic selection of the optimal subset of image pairs leading to the best triangulation performance. The proposed approach is based on two computation stages, the first being the triangulation of points in the image pairs, and the second being the evaluation of the point cloud metrics for each calibration run by preparation and analysis of metric vectors.

The merit of the proposed method is that it goes beyond the traditional performance characteristic of reprojection error, and instead evaluates the performance of stereo triangulation given the estimated intrinsic, extrinsic and stereo rig parameters. This allows for devising metrics that directly evaluate the intended application of a stereo vision system. At the same time, the approach is self-contained in a sense that no additional data collection is made, and the existing pool of image pairs is used as a basis of performance evaluation. This aspect also allows for performing the calibration process in a highly

automated manner, with preserving a transparency of the underlying processing steps and the associated computation results.

The proposed method is validated on a dataset comprised of image pairs of a chessboard calibration pattern gathered with two identical Prosilica GC1350 cameras. The computed metrics and the associated ranking of calibration runs allowed selection of the best-performing result of stereo calibration. Furthermore, because the latter stage of the analysis pipeline results in a dataset similar to those dealt with in unsupervised machine learning, namely a set of feature vectors per calibration run (see Table 2), principal component analysis with subsequent k-means clustering was performed to further highlight the nature of the distribution of the stereo reconstruction performance.

For the further work, it is beneficial to evaluate the proposed method with a concrete stereo vision application use case. In addition to a broader validation of the method, this will allow for uncovering additional application-specific metrics that may be useful in achieving more optimal system calibration.

### Funding
This work was supported by the center for research-based innovation SFI Manufacturing (No. 237900), funded by the Norwegian Research Council. The funders had no role in study design, data collection and analysis, decision to publish, or preparation of the manuscript.

### Grant Disclosures
The following grant information was disclosed by the authors:
Norwegian Research Council: 237900.

### Competing Interests
The authors declare that they have no competing interests.

### Author Contributions
- Oleksandr Semeniuta conceived and designed the experiments, performed the experiments, analyzed the data, performed the computation work, prepared figures and/or tables, authored or reviewed drafts of the paper, and approved the final draft.

### Data Availability
Implementation of the method is available at GitHub under the 3-clause BSD license: https://github.com/semeniuta/vcalib.

The experimental data, along with the Jupyter notebook visualizing the analysis and the code underlying the method implementation, are available at GitHub:
https://github.com/semeniuta/subset-calib-data.

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
