# Peer review of "Subset-based stereo calibration method optimizing triangulation accuracy"

_PeerJ Computer Science, doi:10.7717/peerj-cs.485_

## Round 0.1 · original submission · Major Revisions

The authors must revise this manuscript carefully based on all the comments provided by three reviewers. Then, it can be reviewed again for possible publication.

·

Basic reporting

The paper has been well organized. All literature, figures, and tables support the article. Only a typo as inline 333 (in fig Figure12...) is found.

Experimental design

The research has been designed with many dimensions. The experiment covered the hypothesis.

Validity of the findings

no comment

Additional comments

This paper proposed the method to perform the optimized calibration of the stereo vision system.

Reviewer 2 ·

Basic reporting

This article is very interested that try to propose the optimization for stereo-camera imaging calibration method along with the OpenCV implementation. The explanation is clear through the article. However, in the literature review, the stereo imaging setups are widely used in 3D reconstruction. Is there any application else that required the stereo imaging setups? If any, please explain to give more reason why we need to develop the new stereo calibration optimization method.

Experimental design

This article gives a well-defined research question that try to optimize the setup of the stereo imaging calibration. The experimental design and arrangement is well defined.

The suggest is in figure 4 to give this image gain more understanding, please include the 3 steps: 1) The total set of image pairs, 2) identify corners, and 3) store indices of image pairs that mentioned in line 212-214 along with computational graph for the process of image points preparation.

Validity of the findings

- In the part of "Principle component analysis". Why this article apply only one clustering algorithm that is K-Means? Can we use another clustering algorithm?

- For the experimental result. There is only the result of the proposed method. Is possible to show the result of the comparison of traditional method v.s. the proposed method of stereo calibration optimization.

Additional comments

The figure 1, figure 2, figure 13, and figure 14 are not mentioned in the contexts, please verify.

Reviewer 3 ·

Basic reporting

The author presents an approach to select the best subset of images to calibrate a stereo rig using the well-known library OpenCV for image processing. The author has implemented a routine to try different sets of calibration images and evaluated them in terms of several metrics. By using these metrics the author selects the best subset of images and final calibration.

The article is well written and structured and the literature review is concise and well linked to the publication.

The presented figures are of exceptional quality and help the reader understand the contents fully.

Experimental design

The author validates the approach with a single dataset of 264 stereo images (I expect them to be time-synchronised) and presents several calibration runs together with some metric values in particular cases of the runs.

Please note line 292 Jmdir is said to be an outlier if Jmdir > 25 mm, but figure 8 also suggests an outlier is any run with Jmdir < 15 mm

Looking at Table 2 results, given that the standard deviation of the plane is approx. 0.95 there is no statistical difference between any of the runs. I would suggest the author to validate the results using more precise methods.
In my opinion, there is a lack of information in terms of the geometry used in the experimental design. What is the expected calibration resolution? What is the intended working distance? At which distance was the calibration pattern captured in the images?

There is also another important detail: how does your method ensure that all the image are is covered at some point by a calibration point and avoid skewed or biased distortion models?
Are "good candidates" the points at the centre of the image (less distorted) or are they normally distributed across the image?

Figure 12 mentions a deviation from the nominal. What is the nominal value here? Has the standard deviation also been deviated from the nominal? That is not stated.

I personally do not see the value of the PCA data transformation. Could the author validate that with the reprojection error at some point?

Validity of the findings

I had a look at the images and they do not look sharp. I would encourage the author to use a harder support material for the checkerboard pattern. Cardboard can easily bend and provide a > 1 mm deviation inaccuracy.

The presented results are not statistically sound and controlled. I believe further work needs to be done from data gathering to better analysing it.

---

## Round 0.2 · accepted · Accept

The authors have attempted to respond to all the comments raised by reviewers, thus it is now acceptable. However, please go through the manuscript carefully to fix any possible mistakes, language-wise and presentation-wise.

·

Basic reporting

Approved with the corrections.

Experimental design

no comment

Validity of the findings

no comment

Additional comments

The article is appropriate for publication.

Reviewer 2 ·

Basic reporting

no comment

Experimental design

no comment

Validity of the findings

no cemment

Additional comments

Author revised the paper as suggestion and give clearly answer.

---

## Author Rebuttal · Round 0.2

Dear Dr Boongoen and the reviewers,

First of all, I would like to express my gratitude for your efforts reviewing my paper, "Subset-based stereo calibration method optimizing triangulation accuracy". Your comments and suggestions have helped me to improve the quality of the paper and make my idea clearer to a reader.

In this letter, I would like to provide my responses to your comments. I have highlighted the original comments with yellow background, with my responses provided on the white background.

The main points of improvement in the revised manuscripts are the following:

- Clearer motivation of the proposed method
- Clearer explanation of how the proposes method relates to the traditional planar calibration and why the proposed approach is a viable solution for automating planar stereo calibration
- Analysis of the geometric aspects of the experiment, including the coverage of the image planes by points
- More detailed explanation of accuracy assessment
- Fixed and clarified mathematical expressions
- Improved references to the figures

Best regards,
Oleksandr Semeniuta

## Editor

The authors must revise this manuscript carefully based on all the comments provided by three reviewers. Then, it can be reviewed again for possible publication.

All the reviewer's comments are addressed in the paper and in the responses below.

## Reviewer 1

The paper has been well organized. All literature, figures, and tables support the article. Only a typo as inline 333 (in fig Figure12...) is found.

Thank you for the positive feedback. The typo has been fixed.

The research has been designed with many dimensions. The experiment covered the hypothesis.

Thank you for the positive feedback.

This paper proposed the method to perform the optimized calibration of the stereo vision system.

Thank you for the positive feedback.

## Reviewer 2

This article is very interested that try to propose the optimization for stereo-camera imaging calibration method along with the OpenCV implementation. The explanation is clear through the article. However, in the literature review, the stereo imaging setups are widely used in 3D reconstruction. Is there any application else that required the stereo imaging setups? If any, please explain to give more reason why we need to develop the new stereo calibration optimization method.

The motivation for the method, along with its relation to the state-of-the-art method have been highlighted more explicitly in the Introduction. In a nutshell, the proposed technique does not aim to replace the well-known planar calibration method, but to conduct the latter in an automated manner according to the set of the proposed rules.

This article gives a well-defined research question that try to optimize the setup of the stereo imaging calibration. The experimental design and arrangement is well defined.

Thank you for the positive feedback.

The suggest is in figure 4 to give this image gain more understanding, please include the 3 steps: 1) The total set of image pairs, 2) identify corners, and 3) store indices of image pairs that mentioned in line 212-214 along with computational graph for the process of image points preparation.

Thank you for the suggestion. The proposed reference to Figure 4 has been integrated in the text.

In the part of "Principle component analysis". Why this article apply only one clustering algorithm that is K-Means? Can we use another clustering algorithm?

For the analysis presented in this paper, clustering is used as a tool to automatically label two classes of points that are very clearly distinct. As such, there is no reason to compare different clustering methods.

For the experimental result. There is only the result of the proposed method. Is possible to show the result of the comparison of traditional method v.s. the proposed method of stereo calibration optimization.

Because the proposed approach is not a totally new calibration method, but rather an automated realization of the calibration procedure using the traditional method, it is hard if not impossible to devise a method comparing the two. It is worth noting that, as mentioned in the Introduction, for the traditional method to be more accurate, a number of heuristic

rules have to be applied while collecting the calibration images. However, no one has attempted to describe an automated method aimed at optimizing the calibration results by based on the input set of images or image pairs. As such, the presented paper has a potential to lay the ground for such methods by providing one possible implementation.

In any case, the subsection "Accuracy assessment" presents the efficacy of the proposed method with respect to accuracy of stereo triangulation. This section has been improved in the revised version of the paper to be clearer and more readable.

The figure 1, figure 2, figure 13, and figure 14 are not mentioned in the contexts, please verify.

Thank you very much for pointing this out. All the respective references have been added to the manuscript.

## Reviewer 3

The author presents an approach to select the best subset of images to calibrate a stereo rig using the well-known library OpenCV for image processing. The author has implemented a routine to try different sets of calibration images and evaluated them in terms of several metrics. By using these metrics the author selects the best subset of images and final calibration.

The article is well written and structured and the literature review is concise and well linked to the publication.

Thank you for the positive feedback.

The presented figures are of exceptional quality and help the reader understand the contents fully.

Thank you for the positive feedback.

The author validates the approach with a single dataset of 264 stereo images (I expect them to be time-synchronised) and presents several calibration runs together with some metric values in particular cases of the runs.

You are right. The image pairs are time-synchronized. This is highlighted in the beginning of the "Experimental setup and data collection" section.

Please note line 292 $J_{mdir}$ is said to be an outlier if $J_{mdir} > 25$ mm, but figure 8 also suggests an outlier is any run with $J_{mdir} < 15$ mm

Thank you very much for pointing this up. The formulation in the text has been fixed.

Looking at Table 2 results, given that the standard deviation of the plane is approx. 0.95 there is no statistical difference between any of the runs. I would suggest the author to validate the results using more precise methods.

The standard deviation is computed only for the cases of image pairs that produced acceptable triangulation results. The first rows shown on the table show the same high number of such acceptable cases (a = 257), so they are certainly comparable. However, not all calibration results in the group without degenerate configurations result in the same value of a. As such, the proposed method aims at optimizing accuracy across many different image pairs.

In my opinion, there is a lack of information in terms of the geometry used in the experimental design. What is the expected calibration resolution? What is the intended working distance? At which distance was the calibration pattern captured in the images?

This is a valid question. Overall, the proposed method is aimed at multitude of different applications, and the respective geometries will largely be depended on the lenses used on the stereo setup. In any case, for the studied dataset, a histogram is generated (see Figure 13) that shows how the z-translation of the calibration object with respect to the left camera is distributed. One can see that the operating range of the studied stereo rig is between 0.5 and 1 meters.

There is also another important detail: how does your method ensure that all the image are is covered at some point by a calibration point and avoid skewed or biased distortion models?

Are "good candidates" the points at the centre of the image (less distorted) or are they normally distributed across the image?

This is a very good idea to analyze the coverage of the image space by calibration points. It is not directly applied in the presented method, but can be a great addition to an improved version of the method. I have added a new subsection to the paper ("Experiment assessment"), which includes the analysis of the points' coverage of both the left and the right image plane. Figure 12 includes visualization of the coverage, along with showing how the points of the highest-ranked calibration run are distributed.

Figure 12 mentions a deviation from the nominal. What is the nominal value here? Has the standard deviation also been deviated from the nominal? That is not stated.

Thank you for pointing this up. The details of the calculation are clarified.

I personally do not see the value of the PCA data transformation. Could the author validate that with the reprojection error at some point?

In this paper, PCA in used as an additional visualization tool rather than allows to look the measured metrics in two-dimensional space. The proposed idea to couple PCA with visualization of reprojection error is good. However, as mentioned in the paper,

reprojection error is measures with respect to only one camera, and the proposed method tries to assess triangulation rather than reprojection (the latter is already optimized during calibration of each camera in the individual runs).

I had a look at the images and they do not look sharp. I would encourage the author to use a harder support material for the checkerboard pattern. Cardboard can easily bend and provide a > 1 mm deviation inaccuracy.

The presented results are not statistically sound and controlled. I believe further work needs to be done from data gathering to better analysing it.

This is a very good point, and I totally agree that the calibration object should be as flat as possible with the best possible printing quality. At the same time, as Figure 8 shows, even with the present quality of the calibration object and the captured images, the triangulation accuracy is rather high in most cases. At the present time, unfortunately, there is a limited access to the laboratory due to the COVID-19 situation, so re-running the experiment with a better calibration object is challenging.